# Bamboo-Based Biochar: A Still Too Little-Studied Black Gold and Its Current Applications

Silvana Alfei [1,*] and Omar Ginoble Pandoli [1,2,*]

1 Department of Pharmacy (DIFAR), University of Genoa, Viale Cembrano, 4, 16148 Genoa, Italy
2 Departamento de Química, Pontifícia Universidade Católica, Rua Marquês de São Vincente, 225, Rio de Janeiro 22451-900, Brazil
* Correspondence: alfei@difar.unige.it (S.A.); omar.ginoblepandoli@unige.it or omarpandoli@puc-rio.br (O.G.P.)

**Abstract:** Biochar (BC), also referred to as "black gold", is a carbon heterogeneous material rich in aromatic systems and minerals, preparable by the thermal decomposition of vegetable and animal biomasses in controlled conditions and with clean technology. Due to its adsorption ability and presence of persistent free radicals (PFRs), BC has demonstrated, among other uses, great potential in the removal of environmental organic and inorganic xenobiotics. Bamboo is an evergreen perennial flowering plant characterized by a short five-year growth period, fast harvesting, and large production in many tropical and subtropical countries worldwide, thus representing an attractive, low-cost, eco-friendly, and renewable bioresource for producing BC. Due to their large surface area and increased porosity, the pyrolyzed derivatives of bamboo, including bamboo biochar (BBC) or activated BBC (ABBC), are considered great bio-adsorbent materials for removing heavy metals, as well as organic and inorganic contaminants from wastewater and soil, thus improving plant growth and production yield. Nowadays, the increasing technological applications of BBC and ABBC also include their employment as energy sources, to catalyze chemical reactions, to develop thermoelectrical devices, as 3D solar vapor-generation devices for water desalination, and as efficient photothermal-conversion devices. Anyway, although it has great potential as an alternative biomass to wood to produce BC, thus paving the way for new bio- and circular economy solutions, the study of bamboo-derived biomasses is still in its infancy. In this context, the main scope of this review was to support an increasing production of BBC and ABBC and to stimulate further studies about their possible applications, thus enlarging the current knowledge about these materials and allowing their more rational, safer, and optimized application. To this end, after having provided background concerning BC, its production methods, and its main applications, we have reviewed and discussed the main studies on BBC and ABBC and their applications reported in recent years.

**Keywords:** biochar (BC); pyrolysis; hydrothermal carbonization; bamboo biomass; bamboo-derived BC (BBC); activated BBC (ABBC); environmental xenobiotics removal; circular economy solutions

## 1. Introduction: Biochar (BC)

According to the definition provided by the International Biochar Initiative (IBI), biochar (BC) is "the solid material obtained from the thermochemical conversion of biomass in an oxygen-limited environment" [1]. Practically, BC is the lightweight black residue made mainly of carbon and ashes but also rich in aromatic systems and minerals produced by the thermal decomposition of organic material biomass, including wood, manure, leaves, waste, etc., under limited supply or in absence of oxygen ($O_2$). Such a thermal decomposition process is referred to as pyrolysis. The most commonly used pyrolysis temperature is in the range of 400–600 °C [2], but a larger range of 200–1000 °C is reported [3]. When the thermal decomposition temperature is over 1000 °C and up to 1600 °C, the process is named gasification. Pyrolysis comprises slow pyrolysis, fast pyrolysis, and flash pyrolysis,

depending on the working temperature, the heating rate, and the residence time of the organic matter at a fixed temperature [4]. Upon the pyrolysis of biomass, three types of products with different physical states can form simultaneously in different concentrations. The solid residue comprises BC, which could contain ash and soot, bio-oils represent the liquid product, and the gaseous ones consist of syngas [4]. The relative yields of such products depend mainly on the pyrolysis temperature and heating rate, as shown in Figure 1.

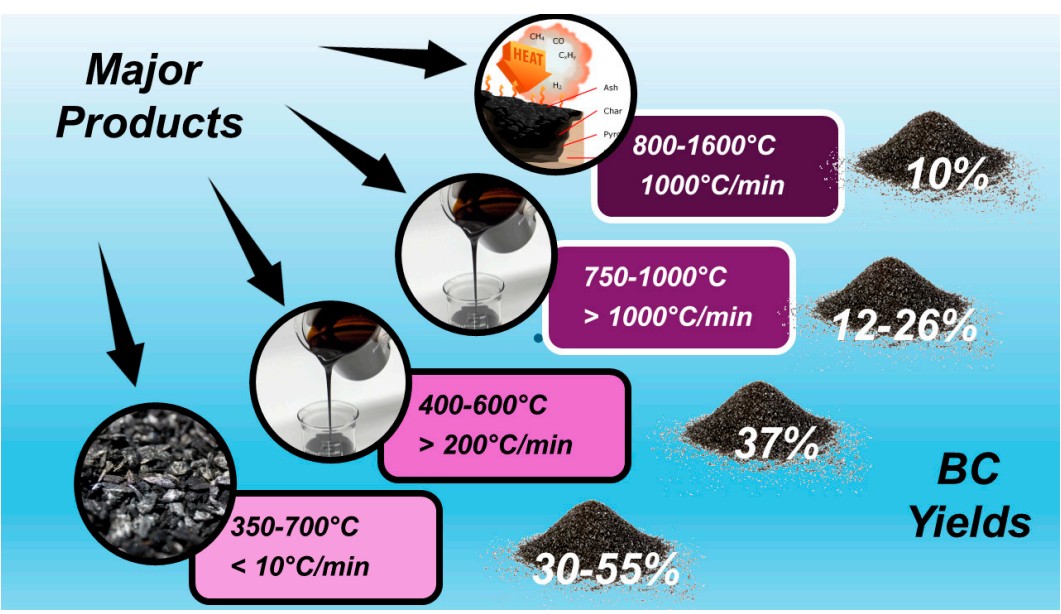

**Figure 1.** Influence of temperature of thermal degradation process on BC yield and that of by-products.

In conditions of slow pyrolysis (350–700 °C, heating rate < 10 °C/min), the major compound is BC; bio-oils are instead the most abundant products in conditions of flash and fast pyrolysis (400–600 °C, heating > 200 °C/min and 750–1000 °C, heating > 1000 °C/min, respectively), and syngas becomes the major product when gasification conditions are applied (800–1600 °C, heating 1000 °C/min) [5–9]. Syngas consists of a gas mixture whose composition can vary significantly depending on the feedstock type and gasification condition. However, typically, syngas is 30–60% carbon monoxide (CO), 25–30% hydrogen ($H_2$), 0–5% methane ($CH_4$), and 5–15% carbon dioxide ($CO_2$), plus a lesser or greater amount of water vapor, smaller amounts of sulfur compounds such as hydrogen sulfide ($H_2S$) and carbonyl sulfide (COS), and finally, some ammonia and other trace contaminants [10]. As for bio-oils, their production could be an appealing strategy for generating fuel from biomass, allowing a reduction in the global dependence on fossil fuels to produce energy [11]. Bio-oil has a higher heating value when compared to its original feedstock, and when burned, it can potentially generate a lower amount of greenhouse gas when compared to fossil fuels [12]. Consequently, even if its most studied application is still as a burning fuel, other applications are emerging due to its composition, as in the case of foams and resins [13]. Moreover, the latter applications emphasize the potential of bio-oil as a source of value-added chemicals [11].

The treatment of biomass at 200–300 °C heating at <50 °C/min is named torrefaction, leading to a torrefied biomass with a high content of BC (69–80%), while hydrothermal carbonization (HTC) in the presence of oxygen, carried out at 400–1000 °C and a very low heating rate (<1 °C/min), leads to the obtainment of hydro char (HC) (50–70% BC), also referred to as HTC material [3]. HC is a solid material distinct from BC due to its production process and properties [14]. Typically, HC has higher H/C ratios and lower aromatic systems than BC, as well as little or no fused aromatic ring structures. The

pyrolytic process leading to BC echoes that which yields charcoal, which is the most ancient industrial technology developed by humankind. Anyway, although very similar and of analogous origin, while charcoal is used mainly as a fuel, the primary application of BC was for soil amendment, aiming at improving soil characteristics and functions, as well as at preventing the natural degradation of biomass to greenhouse gases by sequestrating carbon, thus reducing gas emissions in the atmosphere and mitigating climate change [4]. As mentioned above, BC could contain other solids such as soot and ash. While soot is a secondary pyrogenic carbonaceous material (PCM) identified as a separate component resulting from gas condensation processes, ash typically includes inorganic oxides and carbonates not containing organic carbon. Figure 2 shows the main possible thermal treatment of biomass and the related main products of decomposition.

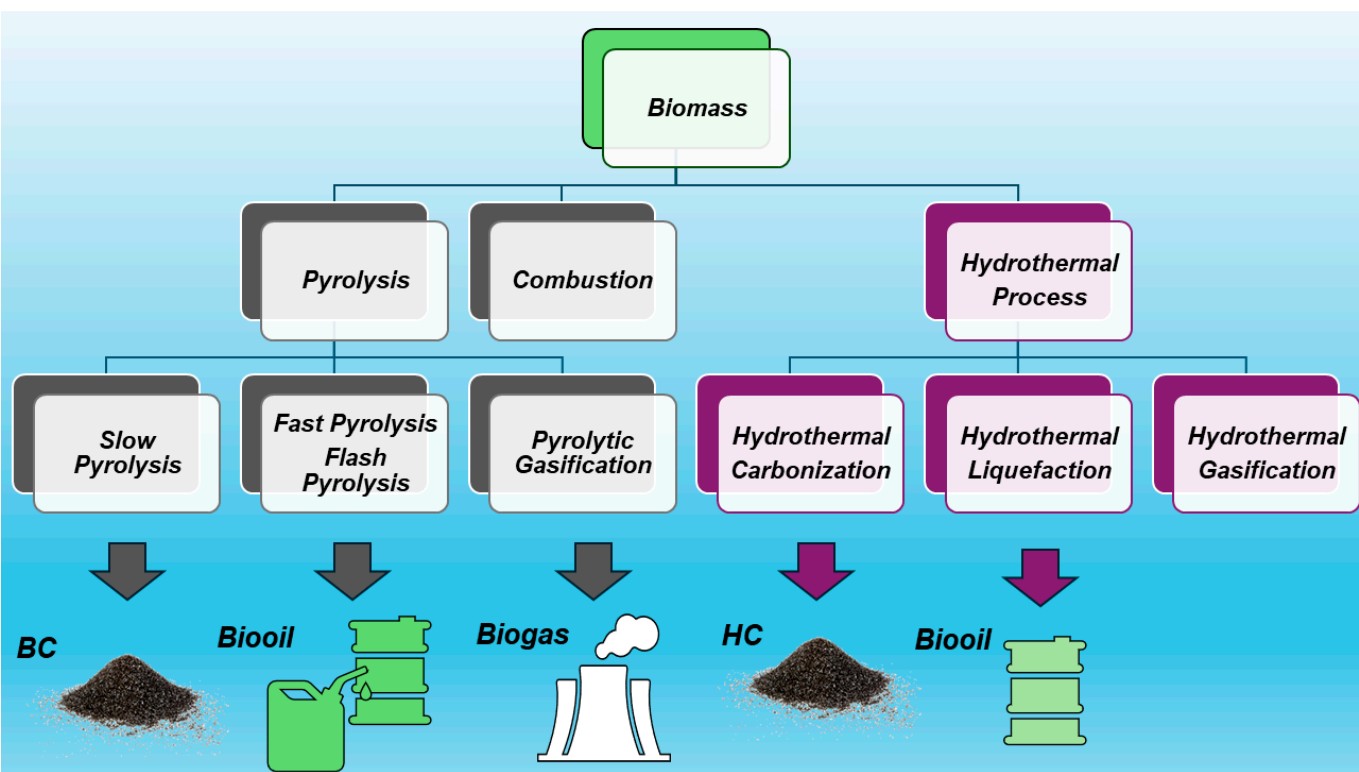

**Figure 2.** Main possible thermal treatment of biomass and the related products of decomposition.

An alternative method to produce BC consists of thermocatalytic depolymerization (TCD), which utilizes microwaves. It has been used to efficiently convert organic matter to BC on an industrial scale, producing $\approx$ 50% char [15,16].

## 2. What Can Biochar Do?

Biochar (BC) is a sustainable, cost-effective, eco-friendly material that is also endowed with reusability, which is increasingly gaining the attention of many researchers [4]. In addition to other possible applications, simply by its production, BC allows for the reduction of agricultural and other types of waste. According to Table 1, among the biomass waste materials, crop residues from agriculture, forestry, municipal solid waste, and food and animal manures have a high potential and are appropriate for BC production [17–22].

**Table 1.** Main sources and general applications of BCs.

| Source Biomass | Ref. | Application | Refs. |
|:---:|:---:|:---:|:---:|
| Crop residue | [23] | | |
| Kitchen waste | [35] | | |
| Forestry | [36] | Carbon sequestration | [24] |
| Agricultural waste | [37] | Soil amendment | [25] |
| Sugar beet tailings | [38] | Composting | [26] |
| Forest residues | [39] | Wastewater treatment | [27] |
| Waste wood | [40] | Concrete additive | [28,29] |
| Bioenergy crops | [41] | Adsorbing xenobiotics | [30] |
| Municipal solid waste | [42] | Reducing greenhouse gas emissions | [31] |
| Wheat straw | [43] | Pollutant degradation | [32] |
| Rice straw | [44] | Catalysis | [33] |
| Food manure | [45] | Stock fodder | [34] |
| Animal manure | [46] | | |
| Corn cob | [47] | | |

BC possesses several properties, including large surface area, high porosity, presence of functional groups, high cation exchange capacity (CEC), high water-holding capacity (WHC), strong stability, etc., which make it suitable for various applications, as reported and summarized in Table 1 [4]. Particularly, in the first column of Table 1 (source biomass), we have reported the biomasses that are commonly used to produce BC along with related references (column two), while in the third column (applications), the most common uses of BCs, which can be derived from all biomasses reported in column one, have been included along with related references (column four).

More specifically, Table 2 reports the environmental applications of BC and the related mechanisms.

**Table 2.** Main environmental applications of BC and related mechanisms.

| Application | Mechanisms | Refs. |
|:---:|:---:|:---:|
| Climate change mitigation | Sequestering carbon in soil, $\Downarrow CO_2$ emissions<br>$\Downarrow NO_2$ emissions, $\Downarrow CH_4$ emissions<br>Tackling 12% of current anthropogenic carbon emissions | [31] |
| Soil improvement | $\Uparrow$ Physicochemical and biological properties<br>$\Uparrow$ Water retention capacity, $\Downarrow$ nutrient leaching<br>$\Downarrow$ Acids, $\Uparrow$ microbial population and microbial activity<br>Positive impacts on earthworm population<br>Preventing desiccation | [25] |
| Waste management | Simply by pyrolyzing waste biomass * | [48] |
| Energy production | By conversion of waste biomass to BC **, providing liquid fuel (bio-oil) | [49] |
| Capturing contaminants | By adsorption of both organic xenobiotics and metal ions present in soil and water | [30,32] |
| Composting | $\Uparrow$ Physicochemical properties of composting<br>$\Uparrow$ Enhance composting microbial activities<br>$\Uparrow$ Organic matter decomposition | [26,32] |

* Including crop residues, forestry waste, animal manure, food processing waste, paper mill waste, municipal solid waste, and sewage sludge; ** mainly by fast pyrolysis; $\Downarrow$ low, lower, reduced, minor; $\Uparrow$ high, higher, increased, major.

By using BC, it is possible to fight global warming by reducing greenhouse gas emissions, such as $CO_2$, $NO_2$, and $CH_4$, and sequestering carbon [31]. Interestingly, BC has been estimated to be capable of tackling 12% of the current anthropogenic carbon emissions [4]. Concurrently, the process of pyrolysis manages to balance fossil fuel consumption by producing clean and renewable energy (bioenergy) [49]. Due to its high content of

carbon, BC can work as a soil conditioner, mainly by improving the physicochemical and biological properties of soils, increasing its WHC by ~18%, reducing nutrient leaching, and neutralizing acidic soils, thus enhancing plant productivity, seed germination, plant growth, and crop yields [25]. Especially when wet BC is applied, soil desiccation is prevented [4]. In some BC systems, all these objectives can be met, while in others, a combination of two or more objectives will be obtained. Interestingly, BC has been used in animal feed for centuries. Doug Pow, a Western Australian farmer, who won the Australian Government Innovation in Agriculture Land Management Award at the 2019 Western Australian Landcare Awards for his studies and innovations, explored the use of BC mixed with molasses as stock fodder for ruminants [50]. The study demonstrated that BC-assisted digestion was improved, and methane production was reduced. On-farm evidence indicated that the fodder led to improvements of liveweight gain in Angus-cross cattle [50]. Additionally, the production of BC by solid waste can help in solving the problem of waste management [48]. The production of the cement known as ordinary Portland cement (OPC), which is an essential component of concrete mix, is energy- and emissions-intensive and accounts for around 8% of global $CO_2$ emissions. To limit this issue, the concrete industry has increasingly moved toward the use of supplementary cementitious materials (SCMs) and additives to both reduce the volume of OPC in the concrete mix and maintain or improve concrete properties [51–53]. BC has been shown to be an effective SCM, reducing concrete production emissions while maintaining the required strength and ductility properties [28,29]. Studies have been carried out demonstrating that a 1–2% wt. concentration of BC is excellent for use in concrete mixes, both in terms of low cost and good strength [28]. Moreover, the addition of a 2% wt. BC solution in the concrete mix resulted in an increased concrete flexural strength by 15% in a three-point bending test conducted after 7 days, compared to the traditional OPC-based concrete [29]. BC-based concrete also showed higher temperature resistance and reduced permeability [28]. Compared to other SCMs from industrial waste streams (such as fly ash and silica fume), BC also demonstrated decreased toxicity. Additionally, BC has been demonstrated to be capable of catalyzing chemical reactions and removing hazardous organic and inorganic pollutants from water soil and atmosphere, including antibiotics, thus reducing the emergence of microbial resistance [4]. Batch and column sorption experiments have shown that certain types of BCs have good sorption performance for heavy metals, dyes, or phosphate from aqueous solution and are being investigated as cost-effective, promising, and eco-friendly alternative materials for producing adsorbent materials [54–60].

### *2.1. BC as Climate and Soil Improver*

As mentioned above, by its carbon sequestration action, BC performs climate remediation.

The refractory stability of BC leads to the concept of pyrogenic carbon capture and storage (PyCCS) [61], or rather, the sequestration of carbon in the form of BC [16]. Due to this potential, BC may be a means of mitigating climate change [62,63]. Due to its alkalinity, BC can increase the soil fertility of acidic soils and improve agricultural productivity [4]. When used for soil application, BC has been demonstrated to improve soil nutrient availability, aeration in soil, and soil water filtration. A large body of peer-reviewed literature exists assessing and describing the crop yield benefits of BC-amended soil [25]. Several field trials using BC conducted in the tropics have shown positive results on crop yields when BC was applied to field soils, and nutrients were managed appropriately. The most well-known example of the beneficial action of BC application to soil is the fertile Terra Preta soils in Brazil, but Japan also has a long tradition of using charcoal in soil [64,65]. Such a tradition is being revived and has been exported over the past 20 years to countries such as Costa Rica. The Brazilian and Japanese traditions together provide long-term evidence of positive BC impact on soils. Figure 3 summarizes the ways by which BC can improve soil quality.

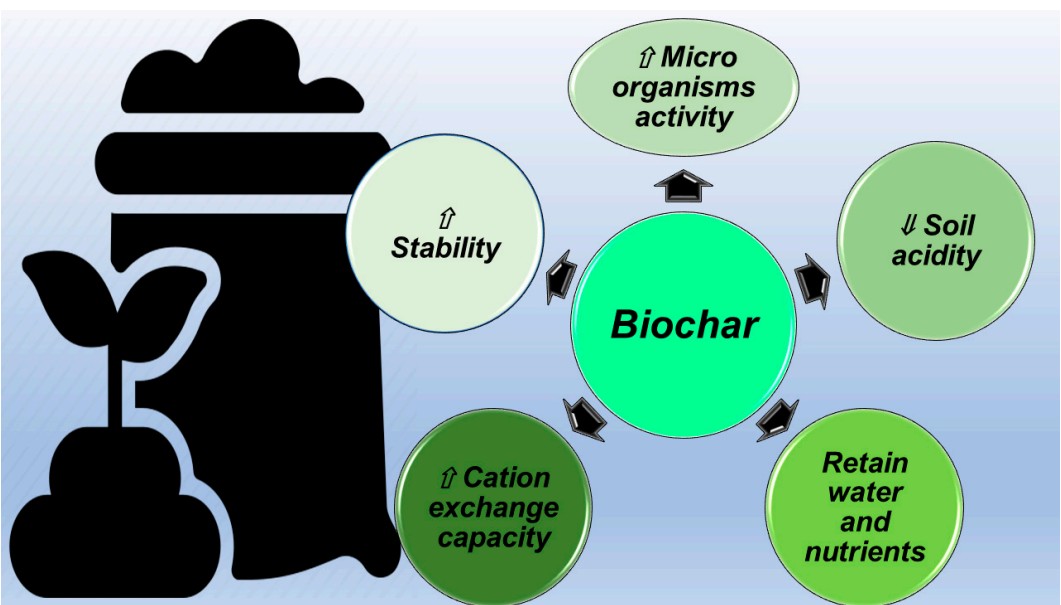

**Figure 3.** Ways by which BC acts as a soil improver. ⇓ low, lower, reduced, minor; ⇑ high, higher, increased, major.

Particularly, BC stimulates the activity of a variety of agriculturally important soil microorganisms by providing them with a suitable habitat and by protecting them from predation and drying while addressing many of their diverse carbon (C), energy, and mineral nutrient needs [66]. BC reduces soil acidity by means of the negative charges present on its surface and attracts and holds soil nutrients by the same mechanism [66]. BC has a high CEC, by which it retains positively charged ions, such as calcium ($Ca^{2+}$), potassium ($K^+$), and magnesium ($Mg^{2+}$), thus improving soil fertility and exchanging them with plant roots. BC mineralizes in soils much more slowly than its uncharred precursor material (feedstock). Moreover, although most BC has a small labile (easily decomposed) fraction of carbon, typically a much larger recalcitrant (stable) fraction composes it [25]. Scientists have shown that the mean residence time (the estimated amount of time that biochar carbon will persist in soils) of this recalcitrant fraction ranges from decades to millennia [25].

The long persistence of BC when incorporated into soils is of paramount importance because it determines how long BC can provide benefits to soil and water quality.

Critical Considerations

While the larger questions concerning overall BC benefits to soils and climate have received affirmative responses, significant questions remain to be answered, including the need for a better understanding of some of the details of BC production and characterization. Although BC is widely viewed as an environmentally positive material for soil, it is crucial to also take into account the potential adverse effects of BC, including disturbing soil pH levels or introducing harmful chemical characteristics that cause problems at the micro dimensions. Therefore, caution should be exercised when considering the extensive application of BC, as the research exploring its positive and negative effects is currently ongoing.

Dust is certainly a concern associated with BC application, but best practices require that BC application be done during periods of low wind to prevent the blowing of fine particulates. Agricultural techniques already exist for applying powdered fertilizers and other amendments. Several techniques are available to help keep wind losses to a minimum, including that BC can be pelleted, prilled, mixed into a slurry with water or other liquids, mixed with manure and/or compost, or banded in rows. The optimization of BC application to soil is important, and the farm technology and methods are available to do the job.

*2.2. Xenobiotics Removal by Biochar (BC)*

BC is a material possessing a large surface area and high porosity based on the conditions of the pyrolysis process and the physicochemical characteristics of the original biomass [4] that allow it to interact with beneficial molecules, including water and water nutrients, as well as hazardous inorganic metal cations and organic pollutants [4]. Due to its enriched porous structure, high surface area, functional groups, and the possibility of interlacing $\pi$–$\pi$ interactions and mineral components, BC is an excellent material for absorbing solutes from aqueous and organic solutions. Moreover, its naturally acquired adsorption capacity (3.6–6.3 g/g for BC prepared at a temperature range of 300–700 °C) [67] can further be enhanced by modifying its physicochemical properties through acid, alkali, or oxidizing treatments, and its surface area can be improved using acid treatments [68–70]. BC can absorb inorganic and organic contaminants, such as polycyclic aromatic hydrocarbons (PAHs), phenols and phthalate derivatives, and antibiotics and other drugs, and can aid in improving the treatment of sewage wastewater containing organic contaminants [71]. Table 3 reports the main mechanisms by which BC captures inorganic or organic pollutants.

**Table 3.** Main mechanisms by which BC can capture inorganic or organic contaminants (reproduced by our recent paper [4]).

| Catching Mechanism | Influencing Factors [#], Details [°], Examples [§] | Ref. |
|---|---|---|
| Adsorption * | $\Uparrow$ Surface area [#]<br>Microporosity of BC [#]<br>pH [#] | [71] |
| Hydrogen bond formation ** | For polar compounds [°],** | |
| Electrostatic attraction/repulsion | For cationic compounds [°]<br>Interaction between positively charged cationic organic contaminants and negatively charged BC surfaces [°],** | |
| Electrostatic outer sphere complexation | Due to metallic exchange with $K^+$ and $Na^+$ available in BC [°],** | |
| Hydrophobic interactions *** | For non-polar compounds [°] | |
| Diffusion | Non-ionic compounds can diffuse into the non-carbonized and carbonized fractions of BC [°] | |
| Formation of surface complexes ** | pH [#]<br>Ionic radius [#]<br>Between metal cations and -OH, -COOH on BCs [°] | |
| Precipitation | Lead precipitates as lead-phosphate-silicate in BC [§]<br>Co-precipitates and inner-sphere complexes can form between metals and organic matter/mineral oxides of BC [§] | |

* From water/soil onto biochar; ** for BCs produced at relatively lower temperatures; *** for BCs produced at higher temperatures; $\Uparrow$ = high, higher, improved, enhanced.

BCs produced at higher temperatures exhibited higher adsorption capacity in the remediation of both organic contaminants and metal cations from soil and water, due to their higher porosity and surface area. Additionally, the adsorption capacity of organic pollutants by BC is superior to that of inorganic ones, which strongly depends on the ionic radius of metals.

Xenobiotics Removal by Degradative Oxidation

The mechanism recognized for years by which BC removes toxic heavy metals and organic xenobiotics from wastewater and contaminated soil is adsorption, whose efficiency mainly depends on the type and number of functional groups, surface area, CEC, etc. Nevertheless, more research studies have evidenced on the surface or inside the BC particles the presence of free radicals, known as permanent free radicals (PFRs) due to their very long lifetime, which may be generated during the pyrolysis of biomass [72]. PFRs can

exist in the air for months or even years. PFR lifetime depends mainly on the type of PFR, their possible complexation with metals or metal oxides, and their carbonaceous structure, as well as their concentration [72]. As recently reported, PFRs' nature depends strongly on pyrolysis conditions, while their formation and characteristics differ based on feedstock types [4]. PFRs are classified into three categories, including oxygen-centered PFRs (OCPFRs), carbon-centered PFRs (CCPFRs), and oxygenated carbon-centered radicals (CCPFRs-O) [4], including, in turn, semiquinone radicals (oxygen-centered), phenoxy radicals (oxygenated carbon-centered radicals), and cyclopentadienyls (carbon-centered radicals) [4]. Several recent studies have mainly investigated the possible contribution of BC-bound PFRs in the removal of several organic xenobiotics through oxidative degradation by electron transfer-dependent production of reactive oxygen species (ROS) [73–83]. The degradation of hazardous cations, including As (III) and Cr (VI), by PFRs via electron transfer has also been extensively reported [84–88], as well as that of biological samples, including hormones and extracellular DNA (eDNA) [89–93]. Odinga et al. reviewed the application of BC-derived PFRs in the remediation of several environmental pollutants [94], while Fang et al. explored the reactivity of PFRs in BC and their catalytic ability to activate persulfate to degrade pollutants [95]. On the other hand, Odinga et al. also considered and commented on the possible ROS-mediated environmental risks of PFRs from BCs, mainly associated with the generation of oxidative stress, which represent the shadows associated with these chemicals and indicate they need further study, more knowledge, and regulation before their extensive application [94].

## 3. Bamboo-Derived Biochar (BBC)

To date, the types of biomasses used to prepare BC and involved in the investigation of BC-bound PFRs mainly include lignocellulosic biomasses, such as pine needles, wheat straw, lignin, waste wood, cow manure, rice husk, and maize straw [23,35–47]. Lignocellulosic biomasses are made of hemicellulose, cellulose, and lignin. Despite the fact that bamboo may be a starting material almost perfect for synthesizing BC (BBC) and activated BBC (ABBC) due to its low cost, high biomass yield, and significantly accelerated growth rate, few researchers and scientists have used bamboo as a source for developing BC, as established by the number of publications on bamboo-derived BCs from the year 2013 until now (351) vs. those on BCs derived from different sources (14560) (Figure 4).

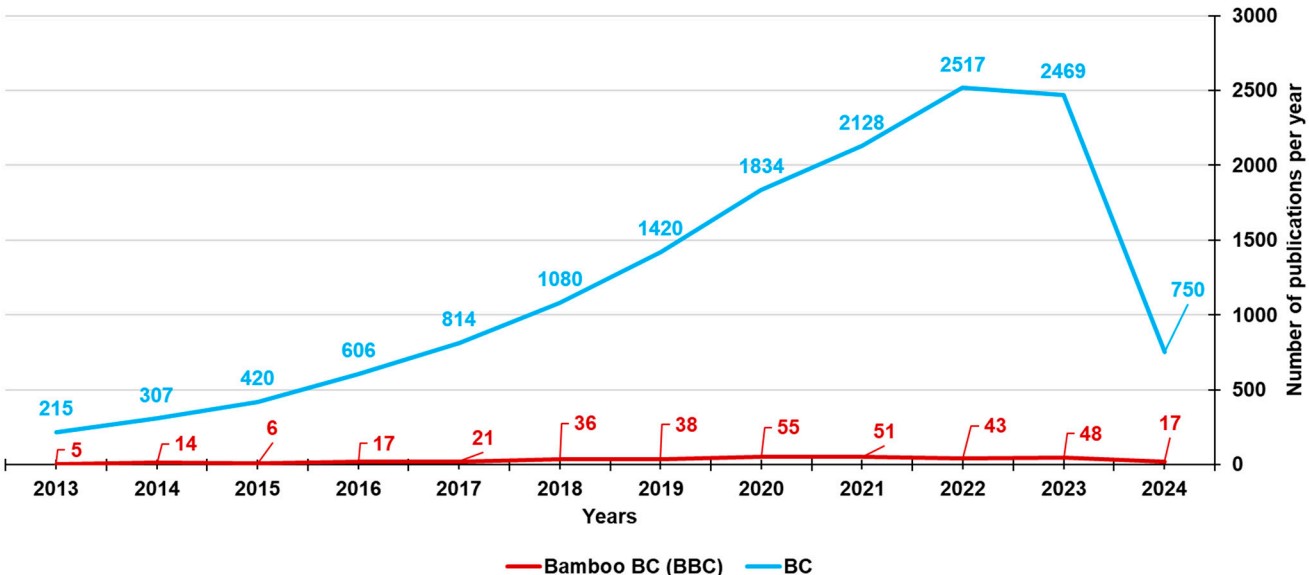

**Figure 4.** Number of papers on BCs and BBCs published from the year 2013 until now, according to the PubMed dataset. The survey was carried out using the keywords biochar (light blue line) and bamboo biochar (red line).

Among the 351 papers on BBC, the majority concerned its adsorption activity (128), followed by its degradation capacity (96), while fewer were studies on its possible toxic action (29) (Figure 5).

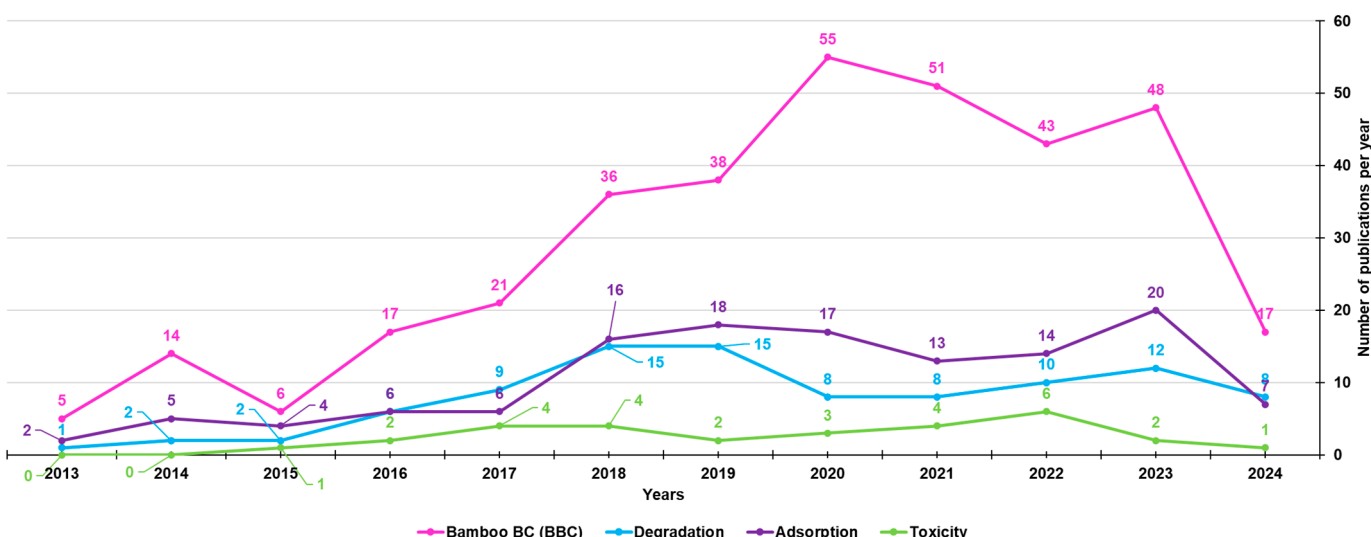

**Figure 5.** Number of papers on BBCs, as well as on BBC absorption, degradation and toxic properties, published from the year 2013 until now, according to the PubMed dataset. The survey was carried out using the keywords bamboo biochar (pink line); bamboo biochar AND adsorption (purple line); bamboo biochar AND degradation (light blue line); bamboo biochar AND toxicity (light green line).

To support and sustain further research on BBCs, the possible associated PFRs, and its possible applications, the following sections of this paper first review the general characteristics of bamboo biomass and the main reported characteristics of BBCs. Secondly, the current applications of BBCs and the PFR-mediated applications of BBCs are reported and discussed.

### 3.1. The Use of Bamboo Biomass to Prepare BC: A Plethora of Advantages over Wood

Bamboo is a woody grass belonging to the Poaceae family, extensively occurring both in forests and in rural areas, as well as in farmlands and riverbanks [96]. Bamboo, due to its growth form, acts as a natural environmental shield and has been used as food and to construct tools and musical instruments. Moreover, bamboo has been exploited as a construction material, as an alternative to wood fuel, as a substitute for wood, and even as a substitute for non-wood forest products [96]. Due to its strength and durability, the latter applications of bamboo are typical in those world areas where the supply of wood no longer manages to meet the demand [97]. Anyway, only in recent years the interest in agricultural bamboo has exploded in Africa, North America, and even Europe. Collectively, as a regenerative agriculture and as a renewable source of energy, bamboo represents a wonder crop for producing BC. It has minimal nutrient needs, it is relatively quick to get established, and once established, it grows with unmatched vigor, using its tenacious roots to raise the water table and curb erosion [96]. Due to these characteristics and mainly thanks to its fast growth with respect to woody plants, bamboo is an optimal source of BC, more efficient and eco-friendlier than wood [97]. Bamboo is capable of proliferating without the necessity of following stringent rules for growing, harvesting, propagating and cutting [98]. Bamboo can tolerate a wide range of temperatures ($-28$–$38\,^{\circ}$C), is able to efficiently exploit water and soil at its disposition, and does not need fertilizers or pesticides. It is known that the possible amount of carbon content in BC deriving from a certain plant depends directly on its metabolic activity (photosynthesis), through which it ingests $CO_2$ and releases oxygen (>30%) back into the atmosphere, thus reducing global warming. In this regard, due to its unequaled growth rate, bamboo yields more biomass and captures more atmospheric

carbon per hectare than anything, thus removing more carbon from the sky and adding it to the soil. Although the usefulness of bamboo as an absorbent is not very well known, that which is cultivated in tropical and subtropical regions worldwide can be used as a renewable feedstock, as already occurs in many countries, especially in Asia (China, India, and Thailand), Central America (Costa Rica, Mexico, and Honduras), and South America (Peru, Ecuador, and Colombia) [99–105]. It has been reported that bamboo plantations can store four times more carbon than timber forests and that they are therefore considered as a potential alternative as a durable carbon stock [106]. Bamboo, like wood, is a promising natural template for biobased devices that take advantage of its hierarchical architecture, microarray channels, and anisotropic mechanical and electrical properties [107]. Bamboo charcoal has been recognized as a cost-effective and environmentally friendly regenerative physio-sorbing material for carbon capture [108]. Additionally, BBC, being a growing porous carbonous material with high surface area, has been used for biomass fuel, carbon capture, and containment adsorption [109]. Through pyrolysis, up to 50% of the carbon can be transferred from plant tissue to the BC, with the remaining 50% used to produce energy and fuels [110]. However, bamboo-derived biochar has not been comprehensively reported despite its good $CO_2$ uptake [111]. Compared to wood composition, the lamellation of bamboo fiber has different fibrillar orientations surrounded by alternating narrow and broad layers [112]. The secondary cell wall is mainly made up of cellulose and lignin, with covalent linkage binding lignin phenolic acids to polysaccharide materials [112]. Bamboo culms grow and mature in a very short period of time, which can enable a continuous supply of fiber, giving it an advantage over trees. Bamboo's mechanical and thermal properties increase its competitiveness against other forms of woody biomasses [113]. For various reasons, bamboo is a superior option to other wood planks to prepare BC, including strength, environmental friendliness, water resistance, cost, soil protection, and contribution to air quality. These benefits were the driving force behind several uses for this material, including blood purification, electromagnetic wave absorbers, and water purifiers. These benefits are influenced by the activation and carbonization procedures used to make bamboo charcoal. Green bamboo wood is roasted to a constant temperature to create activated charcoal when charcoal is exposed to oxygen by using the pyrolysis method.

### 3.2. Composition of Bamboo

The anatomical and chemical components of a plant, especially the plant cell wall thickness and the lignin content, are correlated with its physical and mechanical properties, which can be improved by saturated steam heat treatments [114,115], in turn influencing the characteristics of the derived BC [116,117]. In sight of having to use bamboo to produce BBC with several possible applications, it is important to have more information about bamboo chemical components.

### 3.2.1. Chemical Composition of Bamboo

Table 4 summarizes the main constituents of bamboo.

**Table 4.** Main constituents of bamboo.

| Components | Chemical Structure/Class of Substances | Description | (%) | Refs. |
|---|---|---|---|---|
| Cellulose | Portion of cellulose | Main material responsible for fibers' stability and mechanical strength, allowing the formation of compact fibers<br>Higher thermal stability and resistance to mechanical stresses in comparison to other non-cellulosic plant fiber components<br>Exhibit areas with a flexible structure (amorphous cellulose) and areas with an ordinated, rigid, and non-flexible structure (crystalline cellulose) | 40–60 * | [118–121] |
| Hemicellulose (xylan) ** | The most present hemicellulose in bamboo | Consists of a heterogeneous group of polysaccharides not forming a well-arranged fibrous network (amorphous structure)<br>Low polymerization degree<br>Easily absorbs water | 25 § | [116,122] |

**Table 4.** *Cont.*

| Components | Chemical Structure/Class of Substances | Description | (%) | Refs. |
|---|---|---|---|---|
| Lignin |  Portion of lignin | Amorphous phenolic macromolecule composed of phenyl propane units (C6-C3) No crystalline structure High resistance Cellulose and hemicellulose cells constitute the wall matrix Prevents cellulose and hemicellulose degradation, providing strength and rigidity to plant tissues Energy storage Polymerization degree (PD) > 15,000 High molecular weight | 20–30 | [116,120,121,123] |
| Extractives | N.R. | Aromatic organic compounds, including fatty acids, terpenes, flavonoids, and steroids | N.R. | [120,122,124] |
| Ash | N.R. | Mainly in the interior of the stem | 1–5 | N.R. |
| Starch |  Portion of starch | Production by the cellular activity of chlorophyllized vegetables Attractive for xylophagous organisms, especially for *Dinoderus minutus* | 2–5 | [124] |
| Moisture | N.R. | N.R. | 6.1 | N.R. |
| Proteins | N.R. | N.R. | 1.5–6 | [124] |
| Glucose | N.R. | N.R. | 2 | [124] |
| Waxes, Resins | N.R. | N.R. | 2–3.5 | [124] |
| Silica | N.R. | Mainly in the epidermis, increasing from bottom to top Nonexistent in the internode tissues | 1–6 | N.R. |

* Along with hemicellulose and lignin content, represents more than 90% of the total weight; ** 90% of hemicelluloses in bamboo; § of bamboo cell wall; N.R. Not reported.

The cellulose content in bamboo stem is in the range of 40–60%, while bamboo hemicellulose (25%) is composed of more than 90% xylan (4-O-acetyl-4-O-methyl-D-glucuronoxylan, shown in Table 4), a linear short-chain polymer with a polymerization degree of 200 [116]. Xylans constitute 25% of the bamboo cell wall, which is classified between hardwood and softwood, thus being endowed with a very particular structure [120]. Lignin (20%) occurs in the compound middle lamella as well as in the primary wall structure [123]. Lignin is considered a set of correlated materials responsible for transferring the stress between fibers, which promotes traction resistance, flexibility, and rigidity to the stem longitudinal section, being a supporting element. Lignin exhibits a high molecular weight and is the second most abundant component in bamboo, emerging from several stages of lignification that are related to the plant age. However, bamboo reaches a final height between 3 and 6 months during its growth process. While the process of lignification occurs in the longitudinal direction (in the top-down direction on bamboo length) in the internodes, it occurs in the transversal direction in the inner layer of the region closest to the bark, thus completing the whole process of the growth stage. The lignin and carbohydrate proportion varies during the maturation process and tends to stabilize when the plant is about 1 year old [120]. Secondary elements in bamboo chemical composition are represented by extractives, ash, waxes, resins, and silica. The compounds known as extractives consist of aromatic organic compounds, including fatty acids, terpenes (essential oils), flavonoids, and steroids, that are distributed in the leaf, shell, stem, and other parts of the plant but do not constitute the cell wall. The variation of these secondary components is related to the plant species, age, season, and weather conditions, as well as to the soil nutrient availability and diversity and the presence of water [120,122,124]. Collectively, the content of cellulose, hemicellulose, and lignin in bamboo is similar to that of wood, while the extractives, ashes, and silica contents are higher. The soluble substances (resins, fatty acids, essential oil, tannins, etc.), ashes (inorganic substances such as potassium, calcium, silicon, and magnesium), and lignin (a polymer responsible for the plant elasticity and resistance properties) contents in nodes are lower than in the internodes. For the cellulose content, the opposite occurs [120,121].

3.2.2. Elemental Composition of Bamboo

The elemental composition of a material relates to the average contents of carbon (C), hydrogen (H), nitrogen (N), and oxygen (O), and the H/C and O/C ratios of different bamboo and other lignocellulosic species are given in Table 5.

**Table 5.** Elemental composition of bamboo, wood, and other materials.

| Biomass | C | H | N | O | H/C | O/C | Sources |
|---|---|---|---|---|---|---|---|
| *Bambusa vulgaris* | 49.60 | 6.10 | 0.40 | 44.00 | 0.12 | 0.89 | [125] |
| *Bambusa vulgaris* | 46.80 | 6.38 | 0.22 | 46.60 | 0.14 | 0.99 | [126] |
| *Dendrocalamus giganteus* | 44.26 | 5.48 | 0.46 | 42.66 | 0.12 | 0.96 | [127] |
| *Dendrocalamus latiflorus* | 44.22 | 6.10 | 0.07 | 45.63 | 0.14 | 1.03 | [128] |
| *Phyllostachys makinoi* | 43.90 | 6.06 | 0.06 | 41.47 | 0.14 | 0.94 | [128] |
| *Phyllostachys pubescens* | 45.25 | 5.71 | 0.08 | 43.89 | 0.13 | 0.97 | [128] |
| Wood * | 45.68 | 6.30 | 0.30 | 47.42 | 0.14 | 1.04 | [129] |
| Forest residue | 51.40 | 6.00 | 0.50 | 40.00 | 0.12 | 0.78 | [129] |
| Pine | 47.79 | 5.80 | 0.10 | 45.31 | 0.12 | 0.95 | [129] |
| Rice husk | 47.30 | 6.10 | 0.90 | 45.70 | 0.13 | 0.97 | [125] |
| Sugar cane bagasse | 48.10 | 5.90 | 0.50 | 45.50 | 0.12 | 0.95 | [125] |
| Jatropha bark | 50.80 | 6.50 | 1.50 | 41.30 | 0.13 | 0.81 | [125] |
| Elephant grass | 49.20 | 6.10 | 1.10 | 43.60 | 0.12 | 0.89 | [125] |

C = carbon; H = hydrogen; N = nitrogen; O = oxygen; * Average values for different species.

The biomasses reported in Table 5 show very similar H/C ratios (from 0.12 to 0.14), indicating that they are highly carbonized. Otherwise, the average O/C ratio of bamboo species (0.96) is lower than the O/C ratio of wood biomass (1.04), indicating a higher

hydrophobicity on their surfaces, which makes bamboo biomass more stable against degradation than wood-based biomass. The average oxygen content of the bamboo species (44.04%) is lower than that of wood (47.42%), thus indicating a lower reactivity. Nevertheless, the presence of a high content of oxygen could decrease the energy density as well as the miscibility in hydrocarbon fuels. The biomass nitrogen content ranged from 0.1 to 12%, and the bamboo species had very low nitrogen content that ranged from 0.06 to 0.46%. The lowest content of nitrogen is important for environmental protection since NOx production, when burning the material, promotes environmental damage.

*3.3. Bamboo-Derived Biochar (BBC)*

In the last few years, researchers and scientists have used bamboo biomass from different species as a source for developing BC. Bamboo species like *Dendrocalamus giganteus*, *Dendrocalamus latiforus Munro*, *Dendrocalamus asper, Phyllostachys pubescens Mazel*, *Phyllostachys edulis*, *Phyllostachys virdiglaucesons*, Moso bamboo, and others have been carbonized in different conditions and in several cases chemically activated to obtain activated bamboo biochar (ABBC), which demonstrated different characteristics and properties and was studied for different applications. Table 6 summarizes useful information on the preparation of these materials and their possible uses.

**Table 6.** BBCs and activated BBCs (ABBCs) developed in recent years and their applications.

| Bamboo Biomass | Pyrolysis Conditions | Reactor | Char (%) Other (%) | Characteristics Applications | Refs. |
|---|---|---|---|---|---|
| *Dendrocalamus giganteus* | 300 °C Slow pyrolysis | Fixed bad type | BC 80% ## Oil 35% $ Gas 40% @ | ⇑ Porosity ⇑ Carbon concentration As AC after chemical/physical modification Similar to wood biochar Energy source Soil ameliorant | [127] |
| | 200–1000 °C | Tube furnace (MTI) | N.R. | 700 °C ⇑ Resistivity ⇑ Thermal conductivity ⇑ Thermal heating rate As a 3D microfluidic heater | [130] |
| | | | | 1000 °C ⇑ Electric conductivity As working electrode | |
| *Phyllostachy edulis* | N.R. | N.R. | N.R. | Ag-carbon electrodes for energy device applications | [131] |
| Bamboo waste | $KHCO_3$ 400 °C/3 h | Muffle furnace | N.R. | Excellent electrochemical performance as supercapacitor electrode materials | [132] |
| Bamboo chopsticks | 800 °C/2 h (alkali) | N.R. | N.R. | Sustainable anodes for Li-ion batteries | [133] |
| Bamboo powder waste (alkali-activated) | 1000 °C/15 min | Tube furnace | N.R. | Sustainable anodes for Na-ion batteries | [134] |
| *Dendrocalamus asper* | 400 700 800 900 °C | N.R. | N.R. | N.R. | [135] |
| *Phyllostachys pubescens Mazel* | 900 °C | Tube furnace | N.R. | BCT-derived air cathode for microbial fuel cells | [136] |
| *Phyllostachys edulis* | 350 °C/60 min 500 °C/40 min 900 °C/240 min | N.R. | N.R. | 3D solar vapor-generation device for water desalination | [137] |
| Local defoliated bamboo | Surface-carbonized | N.R. | N.R. | Efficient photothermal-conversion devices | [138] |
| Agricultural by-product (BSS) [d] (*D. latiforus Munro*) | 300 °C to 500 °C | Tubular furnace | 48% | As AC when chemically/physically modified soil ameliorant ⇑ Porosity ⇑ Carbon concentration | [139] |

**Table 6.** *Cont.*

| Bamboo Biomass | Pyrolysis Conditions | Reactor | Char (%) Other (%) | Characteristics Applications | Refs. |
|---|---|---|---|---|---|
| Dry bamboo stalks | 400 °C to 600 °C Slow pyrolysis | Muffle furnace | 32% to 27% | In place of industrially produced AC ⇑ Porosity ⇑ Carbon concentration | [113] |
| Bamboo waste | 500 °C | Fabricated close tank | N.R. | ⇑ Soil fertility and crop growth | [140] |
| Bamboo tick (*P. praecox*) | 700 °C/4 h | Closed container | N.R. | ⇑ Soil acidification ⇑ Soil C and nutrient retention ⇑ Microbial community abundance ⇓ $CO_2$ | [141] |
| Bamboo | 500 °C | N.R. | N.R. | ⇓ $NO_2$ emissions in thermophilic phase of composting ⇑ nosZ-carrying denitrifying bacteria | [142] |
| Commercial BBC | N.R. | N.R. | N.R. | ⇑ Humidification during pig manure composting ⇑ Humic acid (HA) ⇑ HA/Fulvic acid (FA) ratio ⇑ Bacteria transforming organic matter | [143] |
| Residual bamboo biomass | 450–550 °C | N.R. | N.R. | Soil amendment | [144] |
| Bamboo waste | N.R. | N.R. | N.R. | Improving yield of pakchoy plant | [145] |
| Bamboo feedstock | 300–600 °C | Furnace apparatus | N.R. | ⇑ Tomato plant growth ⇑ Fruit quality | [146] |
| Bamboo stems (culms) [a] | 400 °C/30 min | Sealed metallic kiln | N.R. | ⇑ Physicochemical properties of SL, SiL ⇑ Tomato productivity | [147] |
| N.R. | 500 °C/2 h | N.R. | N.R. | ⇓ Cu uptake in roots ⇓ Solubility of soil heavy metals | [148] |
| N.R. | N.R. | N.R. | N.R. | ⇑ pH in red soil ⇑ Soil nutrients ⇑ Abundance of *Basidiomycota Mucoromycota, Chytridiomycota* | [149] |
| N.R. | <500 °C | N.R. | N.R. | ⇓ Mobile Cd, Cu, Mn, Ni, Zn ⇓ Pb, Mn, Cd, Zn, Cu, Ni uptake in soybean shoots ⇑ Root nodulation ⇑ Soybean growth ⇑ Plant K and Mo uptake ⇑ Soybean physiological performance | [150] |

**Table 6.** *Cont.*

| Bamboo Biomass | Pyrolysis Conditions | Reactor | Char (%) Other (%) | Characteristics Applications | Refs. |
|---|---|---|---|---|---|
| Bamboo chips | 300 °C/1 h 450 °C/1 h 600 °C/1 h | Muffle furnace | N.R. | Remediation of As-contaminated paddy soil via iron–organic ligand complexation | [151] |
| Bamboo charcoal particles [b] | 600 °C | N.R. | N.R. | As stabilizer for heavy metals Nitrogen retention in sludge composting | [152] |
| Bamboo sawdust | 1000 °C | N.R. | N.R. | For $CO_2$ capturing ⇓ Regeneration temperature Excellent adsorption capacity | [153] |
| Bamboo carbon | 600 °C/2 h | Muffle furnace | N.R. | BBC [§]-immobilized *Paracoccus* sp. YF1 for nitrates remediation | [154] |
| *P. virdiglaucesons* | 460 °C Slow pyrolysis | Rotary furnace | 50% | Nitrate absorption from wastewater or industrial effluents * | [155] |
| Residual of Moso bamboo manufacturing | 900 °C/1 h | Electric furnace | N.R. | Nitrate-nitrogen adsorption | [156] |
| Giant timber bamboo (*P. bambusoides*) | 400, 700, 1000 °C/1 h | Charcoal kiln | N.R. | Ammonia absorption | [157] |
| Healthy dried stems without leaves of bamboo | 500 °C/20 min | Muffle furnace | N.R. | Modulate the toxic effects of chromium | [158] |
| Offcuts of bamboo [c] | 900 °C/4 h | Vacuum annealing furnace | N.R. | Remediation of Cd (II) in water | [159] |
| Bamboo residues | 400 °C | N.R. | N.R. | Restoration of acidic Cd-contaminated soil | [160] |
| Local bamboo | 600 °C/4 h | N.R. | N.R. | Cu adsorption from soil ⇓ Cu accumulation in lettuce | [161] |
| Bamboo pieces | 600 °C/5 h | Tube furnace | N.R. | Cu absorption from soil ⇓ Soil acidity ⇓ Zn and Pb | [162] |
| Bamboo shoot shells | 500 °C/3 h | Muffle furnace | N.R. | Removal of Ag (I) and Pd (II) Removal of TC and MB | [163] |
| N.R. | N.R. | N.R. | N.R. | Removal of elemental mercury | [164] |
| N.R. °°° | N.R. | N.R. | N.R. | Removal of Cd (II) ions from water | [165] |

**Table 6.** *Cont.*

| Bamboo Biomass | Pyrolysis Conditions | Reactor | Char (%) Other (%) | Characteristics Applications | Refs. |
|---|---|---|---|---|---|
| Makino bamboo (*P. makinoi Hayata*) [d] | 800–900 °C/2 h | Furnace | N.R. | Removal of heavy metal ions from water | [166] |
| Moso (*P. pubescens*) and Ma (*D. latiflorus*) bamboo slices [e] | 800 °C/1 h | Furnace | N.R. | Removal of heavy metal ions from water | [167] |
| *B. vulgaris striata* [f] | 650 °C/2 h | N.R. | N.R. | Adsorption of Cd (II), Hg (II), and Zn (II) from aqueous solution | [168] |
| Bundles of bamboo culms (*Melocanna baccifera*) [g] | N.R./2–4 h | Kiln | N.R. | Removal of Ni (II) and Zn (II) from aqueous solutions | [169] |
| Commercial bamboo charcoal | >450 °C | N.R. | N.R. | Activated by NaOH treatment ⇑ Percentage of surface graphitic carbon ⇑ Oxygen-containing groups ⇑ $\pi$–$\pi$ interactions Adsorptive removal of chloramphenicol | [170] |
| Bamboo pieces | 550 °C | Fluidized bed reactor | N.R. | 100% Furfural removal | [171] |
| Bamboo | 600 °C/1 h | Tube furnace | N.R. | Removal of MB by electrostatic interactions | [172] |
| Bamboo from authors' campus (Jiangsu University, China) | 200 °C/6 h 180 °C/3 h | Teflon-lined stainless steel autoclave | N.R. | As core–shell non-metallic photocatalysts for the photocatalytic decomposition of tetracyclines | [173] |
| Bamboo sawdust | 500 °C | N.R. | N.R. | Removal of fluoroquinolone antibiotics | [174] |
| Bamboo waste | 600 °C | N.R. | N.R. | In situ remediation of PCP | [175] |
| Bamboo waste | 1000 °C | N.R. | N.R. | Removal of MCAB-172 | [176] |
| Bamboo waste | 820 °C | Stainless steel vessel | N.R. | ⇓ Bioavailability of DEP | [177] |
| Moso (*P. pubescens*) bamboo | 800 °C Fast pyrolysis | N.R. | N.R. | Removal of pyridine, indole, quinoline | [178] |
| Bundles of bamboo culms (*Melocanna baccifera*) [h] | 800 °C Fast pyrolysis | N.R. | N.R. | Removal of MB and AO7 | [109] |
| Bamboo dust [i] | 240 °C/2 h | Tube furnace | N.R. | Removal of MB | [179] |
| Bamboo [g] | 700 °C/1 h (1st pyrolysis) 850 °C/2 h (activation) | Tube furnace | N.R. | Removal of MB | [180] |
| Moso (*P. pubescens*) bamboo sections | 700 °C Fast pyrolysis | N.R. | N.R. | Removal of CAF and TC | [181] |

**Table 6.** *Cont.*

| Bamboo Biomass | Pyrolysis Conditions | Reactor | Char (%) Other (%) | Characteristics Applications | Refs. |
|---|---|---|---|---|---|
| Bamboo waste (China) | 650 °C/1 h | Muffle furnace | N.R. | CdSe quantum dots/porous channel BBC for improved photocatalytic degradation of TC | [182] |
| Purchased BBC | N.R. | N.R. | N.R. | Removal of DBT | [183] |
| Bamboo sawdust [g] | 873.15 K/1 h (carbonization) 1073 K/0.5 h (activation) | Tube furnace | N.R. | Removal of NVP | [184] |
| Bamboo waste | 25 °C up to 850 °C | Microwave (2450 MHz) | N.R. | Absorption of toluene and benzene ⇑ Humidity resistance | [185] |
| Bamboo | N.R. | N.R. | N.R. | Extraction and determination of coumarins from *Angelicae pubescentis* Radix | [186] |
| Bamboo wood (KOH-activated) | 500–700 °C/60–120 min | Vacuum pyrolysis machine | 29–34% char 37–39% oil 26–33% gas | Adsorption of $CO_2$ and $PM_{2.5}$ | [187] |
| Bamboo power * | 450–600 °C | 2 L Cylindrical reactor | 9–29.82% ** 0.91–2.41% *** | As BC-supported sulfonic acid catalyst for cellulose hydrolysis | [188] |
| *Phyllostachys edulis* | 353 K/3 h | N.R. | N.R. | BBC sulfonic acid bearing polyamide for microwave-assisted hydrolysis of cellulose | [189] |
| Bamboo waste | 600 °C/30 min Fast pyrolysis | Fixed-bed system | 84.7 wt% °° | To prepare phenols | [190] |
| Bamboo waste # | 600 °C/30 min | Fixed-bed reactor | Char 19% Oil 42% $H_2O$ 18% Gas 18% | Formation of aromatics and phenols | [191] |
| Bamboo waste ° | 600 °C/30 min | Fixed-bed reactor | N.R. | Formation of phenols (67%) | [192] |
| Bamboo activated carbon (BAC) | N.R. | N.R. | N.R. | Sulfonated BAC-based catalyst for oleic acid esterification | [193] |

* From a bamboo processing factory (Zhejiang province, China) pretreated with alkaline carbonates; ** char; *** ash; °°° bamboo charcoal was provided by the Department of Material Science and Engineering, Tsinghua University; HTC = hydrothermal carbonization; § BBC improved the grow of immobilized bacterial cells; # catalytic deoxygenation co-pyrolysis of bamboo wastes and microalgae with biochar catalyst; ° pyrolyzed in absence or presence of activated BBC catalysts (activation with KOH, $K_2CO_3$, $KHCO_3$, or $CH_3COOK$ at 800 °C/30 min); °° carbon content; ## 300 °C; $ over 500 °C; @ 600 °C; [a] *Guadua Angustifolia Kunth*; [b] Yaoshi Charcoal Production Company, Hangz; [c] *Phyllostachys heterocycla* (Carr.), *Mitford cv. Pubescens*; [d] $CO_2$- or water steam-activated; [e] water-activated; [f] water steam-activated; [g] raw or KOH-activated BBC; [h] not-activated and microwave-activated ABBC; [i] ammonium persulfate- and potassium persulfate-pretreated bamboo and then activated by cold alkali; BSS = bamboo shoot shell; ⇑ = high, higher, improved, enhanced; the symbol = means similar, like, comparable; BCT = bamboo charcoal tube; ⇓ = low, lower, reduced, decreased; DBT = dibenzo thiophen; CAF = chloramphenicol; TC = tetracycline; MB = methylene blue; PCP = pentachlorophenol; DEP = diethyl phthalate; MCAB-172 = metal-complex dye acid black 172; NVP = N-vinyl pyrrolidone; SL = sandy loam; SiL = silt loam; $PM_{2.5}$ = fine dust; N.R. = not reported.

The studies reported in Table 6 are related to the transformation of the lignocellulose bamboo biomass into pyrolyzed bamboo BCs (BBCs) in a nitrogen atmosphere. In several cases, BBCs were chemically treated with alkali, acid, and oxidant solutions to increase their porosity and add new chemical functional groups, thus achieving activated BBC, namely ABBC. In the studies, BBC and ABBC were used for metal removal in an aqueous solution or gas phase for ameliorating soil quality by reducing acidity and absorbing heavy metals, as catalysts for organic chemical reactions, as energy sources, and as thermoelectrical devices, as well as for the adsorption of organic compounds, such as dyes, drugs, and aromatic compounds. To limit the problem of deforestation and to meet the pressure of avoiding the use of native forest resources for the production of BC, a woody bamboo (species *Dendrocalamus giganteus Munro*) was pyrolyzed by Hernandez-Mena et al. as a renewable material, and the properties of bamboo-derived BC (BBC) were studied [127]. To this end, the fast-growth bamboo was slow-pyrolyzed in a fixed-bed reactor at temperatures ranging from 300 to 600 °C and at a 10 °C/min heating rate [127]. The obtained BC was analyzed by different analytical techniques, and according to the reported results, the pyrolysis of bamboo yielded both biochar and bio-oil, thus representing a low-cost and sustainable strategy for producing energy and for agricultural applications. Using different bamboo species, including *Dendrocalamus giganteus*, *Phyllostachys edulis*, *D. asper*, *P. pubescens Mazel*, and *P. edulis*, and different parts of the plant as bamboo chopsticks and defoliated bamboo and even bamboo waste, BBCs were produced with electrical, electrochemical, and thermal properties [130–133,135–138]. Particularly, depending on the temperature of the pyrolysis process (700 °C), BBCs demonstrated high resistivity, thermal conductivity, and thermal heating rate, being suitable as a 3D microfluidic heater. Otherwise, BBCs obtained at 1000 °C proved to have high electric conductivity, thus being suitable as working electrodes [130]. Also, other bamboo-derived BCs were proposed as Ag-carbon electrodes for energy device applications [131]. BBCs obtained in specific conditions demonstrated excellent electrochemical performance [132] and were usable as supercapacitor electrode materials [132], as sustainable anodes for Li-ion batteries [133], as anodes for Na-ion batteries [134], as BCT-derived air cathodes for microbial fuel cells [136], as 3D solar vapor-generation devices for water desalination [137], and as efficient photothermal conversion devices [138]. The shoot shell of *D. latiflorus Munro* was used to produce BBCs at temperatures ranging from 300 °C to 500 °C. BBC derived at 400 °C demonstrated the best properties in soil improvement in terms of water-holding capacity (WHC), bulk density, iodine sorption, and metal adsorption. Particularly, soil adsorption capacities of Pb (II), Cr (III), and Cd (II) were improved by approximately 27%, 21%, and 29% with the addition of 2.5% wt. BBC [139]. Aiming at preparing BBCs to be used as soil improvers again, Sahoo and co-workers characterized and compared BCs obtained by pyrolysis at different temperatures (400, 500, and 600 °C) of bamboo biomass (BBCs) with those achieved by pyrolyzing pigeon pea stalk under the same conditions [113]. Briefly, the BC yield, which decreased with increasing temperature, was higher in bamboo than in pigeon pea stalk biomass due to more lignin content and low volatile matter in the former. Both BCs were shown to be highly carbonized and hydrophobic and to have low volatile matter, high porosity, and high fixed carbon, as well as proved to be beneficial for agricultural application, as they had high ash recovery [113]. The pyrolysis temperature of 600 °C was the most effective one in terms of surface area, total pore volume, and high mass fraction of carbon and fixed carbon for both the biomass materials [113]. Other authors, by pyrolyzing bamboo waste, bamboo tick, residual bamboo biomass, and bamboo agricultural by-products at temperatures ranging from 300 °C to 700 °C, prepared BBCs that were successful in enhancing soil fertility, crop growth [140], and soil acidification. Such BBCs have been demonstrated to be efficient in improving soil carbon content, promoting nutrient retention, improving microbial community abundance, and reducing $CO_2$ emissions [141], thus being suitable for soil amendment [144]. Yanan et al. prepared three types of biochar (corn stalk biochar (CSB), rape straw biochar (RSB), and bamboo charcoal (BC)) to investigate the relationship between $N_2O$ emissions from composting

and denitrifying bacterial communities on compost and BC particles [142]. The results showed that $N_2O$ emissions rates were higher in the thermophilic phase of composting and that the average emissions rate was lower upon BC treatment than after treatment with CSB and RSB [142]. The nosZ-carrying denitrifying bacterial community played a key role in reducing $N_2O$ emissions, and the study indicated that BC enhanced the efficiency of $N_2O$ emission reduction by enhancing the abundance of these key genera [142]. In another recent study by the same authors, the effects of bamboo charcoal (BC) and wheat straw biochar (WSB) on the humic acid (HA) and fulvic acid (FA) contents during pig manure composting were investigated [143]. The results evidenced that BC enhanced humification more than WSB and significantly increased the HA content and HA/FA ratio [143]. As in the study previously reported, the bacterial community structure under BC treatment differed from those under the other treatments. Particularly, BC increased the abundance of bacteria associated with the transformation of organic matter, thus influencing HA and FA concentration and improving the humification process during composting [143]. By the application of BBCs to soil obtained by the combustion of bamboo feedstocks, the quality and performance of different plants were enhanced. Particularly, the growth of mustard plants [194] and the yield of pakchoy plants [145], as well as the growth of tomato plants and the quality of their fruits, were improved [146,147]. Villagra-Mendoza et al. reported that the physicochemical changes that occurred in loam soils amended with BBC obtained by the pyrolysis of bamboo stems at 400 °C determined a significant improvement in tomato production yield [147]. Wang et al. demonstrated that the application of 5% bamboo biochar to soil reduced Cu uptake in the roots of Moso bamboo (*Phyllostachys pubescens*) and ameliorated soil physical properties and heavy metal solubility in soil [148]. Applications of biochar have significantly reduced the solubility of soil heavy metals. Authors evidenced that the application of BBC to soils ameliorated the quality of soils in terms of improved pH, nutrients, and abundance of beneficial fungi of *Basidiomycota*, *Mucoromycota*, and *Chytridiomycota* species [149], as well as reduced mobile Cd, Cu, Mn, Ni, Zn, reduced Pb, Mn, Cd, Zn, Cu, and Ni uptake in soybean shoots, and increased root nodulation, soybean growth, K and Mo uptake, and physiological performance of soybean plants [150]. Very recently, Tang et al. prepared BBCs at different pyrolysis temperatures (BB300, BB450, and BB600), which were successful in reducing As availability in paddy soil by adsorption and promoted the complexation of HCl-extractable Fe (III)/(II) and the formation of amorphous iron oxides, which, in turn, facilitated the formation of ternary complex (As-Fe-DOM) with highly stability [151]. Moreover, the abundance of *Geobacteraceae* and *Xanthomonadaceae*, which are common electroactive bacteria, was promoted in the BB450-treated paddy soil, which assisted in forming amorphous iron oxides [151]. Also, Hua et al. evidenced that the incorporation of bamboo charcoal particles prepared at 600 °C into the sludge composting material significantly reduced nitrogen loss (64.1%) and lessened the mobility of Cu and Zn [152]. Furthermore, a BBC with high adsorption capacity for $CO_2$ capture and for low-temperature heat utilization was developed by Ji et al. by pyrolyzing bamboo sawdust at 1000 °C [153]. BBCs were also used for nitrate remediation and for the removal/degradation of hazardous inorganic pollutants. Particularly, Liu et al. immobilized a *Paracoccus* sp. strain YF1 on bamboo carbon obtained at 600 °C to be assessed for denitrification. The results showed that denitrification was significantly improved using immobilized bacteria compared to that of free cells, where denitrification time decreased from 24 h (free cells) to 15 h (immobilized cells) [154]. Aiming at further improving the sorption characteristics of BBC in the denitrification process, a BBC/montmorillonite composite was produced in an experimental pyrolysis reactor using bamboo (*P. virdiglaucesons*) as biomass feedstock by Viglašová et al. This composite was successful in removing nitrates from aqueous solutions, being 1.8-fold more efficient than BBC without montmorillonite [155]. In the past, the efficient removal of nitrogen in the form of nitrate and ammonia using BBCs obtained by pyrolyzing Moso and Giant timber bamboo at temperatures in the range of 400–1000 °C in a tube furnace and charcoal kiln, respectively, have been reported by Mizuta et al. [156] and Asada et al. [157]. Particularly,

the BBC prepared in the study by Asaka et al. was more efficient than commercial activated carbon in removing ammonia from water solution, and its performance improved further upon acidic treatment [157]. Concerning the environmental removal of hazardous metals, healthy dried stems without leaves of bamboo were converted in BBC in a muffle furnace at 500 °C to modulate the toxic effects of chromium on wheat plants [158], while offcuts of bamboo and bamboo residues were pyrolyzed at 900 °C and 400 °C, respectively, obtaining BBCs that were successful in the remediation of Cd (II) in Cd (II)-contaminated water and in the restoration of acidic Cd-contaminated soil [159,160]. Very recently, Ma et al. prepared BBCs and magnetic BBCs by the pyrolysis of bamboo at 600 °C and applied them for copper (Cu) immobilization in agricultural lands [161]. Particularly, magnetic BBC increased the maximum adsorption capacity and mitigated copper accumulation in lettuce. The major adsorption mechanisms of magnetic BBC were chemical precipitation, ion exchange, and metal–$\pi$ complexation. Soil organic matter (SOM) facilitated the immobilization of activated copper [161]. In the same year, Zhang et al. prepared a ferrate-modified BBC and tested it as a composite system for soil amendment [162]. Particularly, the capacity of Fe-modified BBC to absorb heavy metal Cu and to reduce soil acidity was investigated [162]. An absorption efficiency of 276.12 mg/g due to single-layer surface adsorption and chemisorption processes was evidenced. Pore diffusion, electrostatic interaction, and surface interaction were the other possible mechanisms of Fe–BBC interaction with $Cu^{2+}$ ions [162]. Multilayer self-assembled multifunctional bamboo shoot shell biochar microspheres (BSSBMs) showed a wide range of adsorption capacities, demonstrating the capability of absorbing heavy metals such as Ag (I) and Pd (II), antibiotics such as TC, and dyes such as MB from wastewater [163]. The maximum adsorption amounts of BSSBMs on Pd (II), Ag (I), TC, and MB were 417.3 mg/g, 222.5 mg/g, 97.2 mg/g, and 42.9 mg/g, respectively [163]. Moreover, Tan et al. investigated the adsorptive potential of a bamboo-derived BC (BBC) and of modified BBC using $H_2O_2$ for elemental mercury removal [164]. The results evidenced that BBC materials have excellent adsorption potential for elemental mercury, especially after being modified by $H_2O_2$ [164]. Batch adsorption experiments were conducted by Wang et al. to investigate the adsorption capacity of Cd (II) ions from aqueous solutions by BBC [165]. The results showed that the adsorption of Cd (II)ions was very fast initially, and equilibrium was reached after 6 h with an adsorbed capacity of 18 mg/g at pH = 8. Similarly, the batch adsorption capacity of heavy metal ions by $CO_2$ and water steam-activated or water-activated ABBC was investigated by Wang et al. [166] and by Sheng-Fong Lo et al., respectively [167]. The optimum pH values for the adsorption capacity of heavy metal ions were 5.81–9.82 by bamboo-activated carbons. The optimum soaking time was 2–4 h for $Pb^{2+}$, 4–8 h for $Cu^{2+}$ and $Cd^{2+}$, 4 h for $Cr^{3+}$ by *Moso* ABBC, and 1 h for the tested heavy metal ions by Makino ABBC. However, the removal efficiency of heavy metal ions by the tested ABBCs decreased in the order $Pb^{2+}$ > $Cu^{2+}$ > $Cr^{3+}$ > $Cd^{2+}$. Specifically, depending on their interaction with anionic functional groups on the surface of activated ABBC, the adsorption capacity varies from 0.68 to 0.19 mg/g [166,167]. Moreover, water steam-activated mesoporous ABBC from the bamboo species *Bambusa vulgaris striata* achieved high efficiency in the removal of Cd (II), Hg (II), and Zn (II) ions from water solutions [168]. The batch studies suggested the highest adsorption capacities for an activated carbon dose of 0.6 g/L, solution pH = 9 and an equilibrium time of 16 h in static conditions were 240, 248, and 254 mg/g of cadmium, mercury, and zinc, respectively [168]. Other studies using KOH-activated bamboo derivates obtained from *Melocanna baccifera* species were mentioned with lower absorption capacities of 40.5 mg/g of Zn (II) [169]. A number of studies were concerned with the use of commercial bamboo charcoal, as well as that of BBCs prepared by the combustion of bamboo pieces, bamboo sawdust, or bamboo waste at different temperatures (450–1000 °C) in the remediation of hazardous organic compounds, including antibiotics and dyes. Particularly, an NaOH-activated BBC with a high percentage of surface graphitic carbon, oxygen-containing groups, and $\pi$–$\pi$ interactions was used for the adsorptive removal of chloramphenicol [170]. The 100% removal of furfural, as well as the capability of removing MB by electrostatic interac-

tions and decomposing tetracyclines by a photocatalytic process, was demonstrated by BBCs [171–173]. Additionally, the removal of fluoroquinolone antibiotics and MCAB-172, the in situ remediation of PCP, and the reduction of the bioavailability of DEP in soil by BBC applications has also been reported [174–177]. Furthermore, Liao et al. reported the production of not-activated and microwave-activated ABBCs starting from Moso bamboo as biomass feedstock [109,178]. BBCs were obtained by pre-carbonization at 150–270 °C for 2–3 days, carbonization at 270–450 °C for 1–2 days, calcination between 450 and 800 °C for 0.5–1 day, and finally, maintenance at 800 °C for a short time. All the processes were carried out under a nitrogen atmosphere. The obtained materials were successful in the removal of nitrogen-heterocyclic compounds (NHCs), such as pyridine, quinoline, and indole [178], as well as MB and OA7 dyes [109] from water solutions. In the first case, the spent BBC with NHC adsorption was effectively regenerated by MW radiation [178], while microwave-activated ABBCs were remarkably more efficient in removing dyes than the raw BBC (17.32 mg/g vs. 14.99 mg/g for MB and 9.28 mg/g vs. 4.91 mg/g for OA7) [109]. Anyway, a very high maximum monolayer adsorption capacity of 454.2 mg/g for MB was reported previously by Hameed et al. using a KOH-activated ABBC prepared by pyrolyzing in a two-phase process: bamboo biomass first at 700 °C for 1 h and then at 850 °C for 2 h in the presence of KOH [180]. In a recent study, two BBCs were prepared from the torrefaction of ammonium persulfate- and potassium persulfate-pretreated bamboo and then activated by cold alkali, which were named ASBC and KSBC, respectively. The two BBCs demonstrated high adsorption properties (475/881 mg/g at 303 K) over MB, mainly by electrostatic interactions [179]. As previously reported by Fan et al., Liao and colleagues succeeded in removing chloramphenicol (CAF) from water solutions using a BBC obtained from the thermal decomposition of sections of bamboo (Moso bamboo) [181]. Particularly, 4-year-old Moso bamboo cut into different sections was pre-carbonized at 150–250 °C for 1–2 d, carbonized at 250–400 °C for 0.5–1 d, calcined at 400–700 °C for 0.5–1 d, and finally, kept at 700 °C for a short time [180]. In addition to CAF, the obtained BBC was also capable of removing tetracyclines (TCs), both in batch experiments and in fixed-bed column ones [181]. A simple in situ method was used by Men et al. to load CdSe quantum dots (QDs) onto hydrothermal biochar (HTC) obtained by pyrolyzing bamboo to form CdSe/HTC composites [182]. The authors employed the as-obtained BBC-based nanomaterials in the photocatalytic degradation of tetracycline (TC). Compared with pure CdSe quantum dots, the best photocatalytic degradation efficiency of CdSe/HTC complex containing 15% HTC was 73%, which was attributed to the high carrier transport efficiency of HTC and the inhibition reorganization of photogenerated electron-hole pairs [182]. Furthermore, DBT was removed by n-octane by Zhao et al. using a commercial BBC [183], while Li and Umereweneza prepared an ABBC using bamboo sawdust that was carbonized at 873.15 K for 1 h and then activated with alkaline treatment (KOH) at 1073 K for 0.5 h [184]. The as-prepared ABBC was successful in removing NVP from water solutions [184]. The enhanced adsorption capacity towards aromatic volatile organic compounds (VOCs) by hydrophobic porous BBC produced via microwave rapid pyrolysis of bamboo in the presence of $FeCl_3$ was recently reported by Junhao et al. [185]. Particularly, the as-obtained engineered BBC displayed better benzene and toluene adsorption capacity and humidity resistance. A BBC was also employed by Yu et al. to prepare a composite monolithic adsorbent [186]. The prepared composite adsorbent was successfully applied in a column for the reversible absorption of two coumarins from *Angelicae pubescentis* Radix, as well as for their determination [186]. Activated BBC (ABBC) was produced by treating the BBC with KOH, achieving a highly porous structure that was effective in extracting organic and inorganic contaminants from the air [187]. Particularly, ABBC produced at 700 °C demonstrated an air purifier's performance efficiency for removing $CO_2$ and $PM_{2.5}$ of 91.23 and 89.19%, respectively [187]. Bamboo-derived BCs were also used as hydrolytic catalysts or as catalysts to produce aromatic and phenolic derivatives. Particularly, the combustion by the molten alkali carbonate method of bamboo power at 450–600 °C (450 °C being the optimal temperature) in a 2 L cylindrical reactor yielded a BBC that was used to support

sulfonic acid groups, achieving a BBC-supported sulfonic acid catalyst that performed well in cellulose hydrolysis [188]. Similarly, a bamboo-based biochar sulfonic acid (BCSA) catalyst bearing polyamide moieties (BCSA-PA) with acidic and alkaline sites was prepared by the Yin group; it showed high activity and good selectivity for the conversion of glucose to 5-hydroxymethylfurfural (HMF) in pure water at 90 °C under microwave irradiation [189]. Chen et al. used N-doped BC catalysts derived by the fast pyrolysis (30 min) of bamboo at 600 °C in the presence of variant $NH_3$ concentrations for the catalytic pyrolysis of bamboo wastes to prepare phenols [190]. High-concentration aromatic and phenolic derivatives were prepared by Chen et al. through the catalytic deoxygenation co-pyrolysis of bamboo wastes and microalgae using a BBC catalyst, in turn achieved by bamboo waste pyrolysis [191]. Similarly, Yang et al. produced high-concentration phenols (67%) through the catalytic fast pyrolysis of bamboo wastes at 600 °C using a BBC catalyst activated with KOH, $K_2CO_3$, $KHCO_3$, or $CH_3COOK$ at 800 °C, in turn achieved by bamboo waste pyrolysis at 600 °C [192]. Also, Niu et al. prepared a sulfonated heterogeneous acid catalyst from bamboo-activated carbon (BAC) through arylation. This BBC-based catalyst was used in the catalytic esterification of oleic acid with ethanol to produce biodiesel with 96% efficiency [193]. Collectively, information reported in Table 6 has evidenced that the majority of case studies on BBC and ABBC were concerned with their use in the removal of organic and inorganic xenobiotics from wastewater and in soil amendment, followed by their application in developing electrical and photochemical devices. Otherwise, their employment as catalysts for organic chemical reactions remains limited (Figure 6). In fact, unlike BC-based materials obtained from other biomasses, which have been employed as heterogeneous catalysts for organic reactions, including reduction, oxidations, esterification, C-C and C-N coupling, alkylation, epoxidation, cycloadditions, and multi-component reactions [195], the reactions catalyzed by BBCs include only those reported in Table 6, such as cellulose hydrolysis, pyrolytic production of phenols and aromatic compounds, and esterification reactions [188–193].

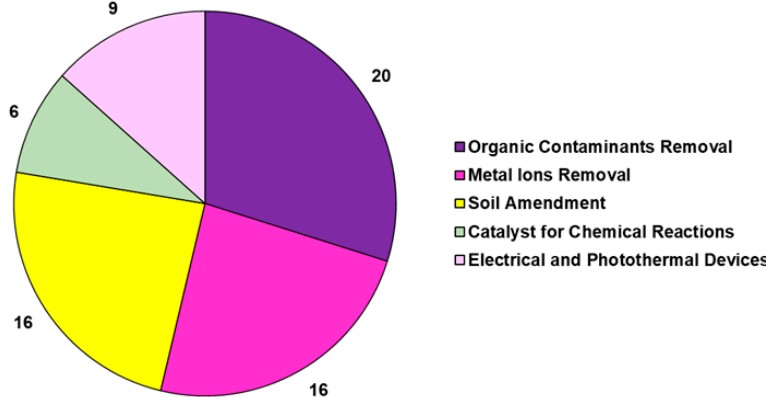

**Figure 6.** Main relative applications of bamboo-derived BC. Numbers refer to the existing publications reporting that specific application.

## 4. BBC-Derived Persistent Free Radicals

While the formation and presence of PFRs in the BC produced by several feedstock biomasses has been widely documented and studied since 2014, the literature documentation regarding those found in bamboo-derived biochar (BBC) is limited to four recent publications (Figure 7), which are reported in Table 7.

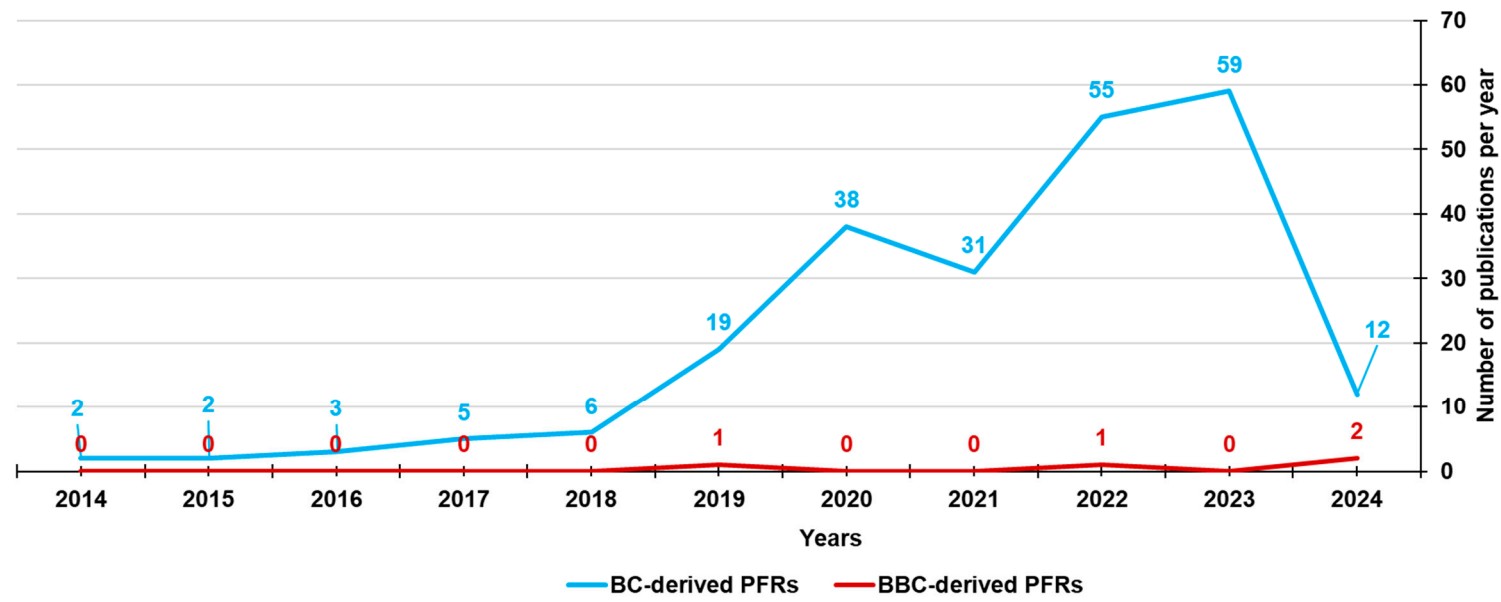

**Figure 7.** Number of papers on BC-derived PFRs and BBC-derived PFRs from 2014 until now, according to Scopus and PubMed datasets (reviews included). The survey was carried out using the following keywords: persistent AND free AND radicals AND biochar (light blue line) and persistent AND free AND radicals AND biochar AND bamboo (red line).

**Table 7.** Bamboo-BC (BBC)-derived EPFRs and their applications.

| Source of BBC | Application | Process | Radicals Active Site | | Radicals | Refs. |
|---|---|---|---|---|---|---|
| Bamboo | Tetracycline degradation | Fenton-like | PFRs | | ●OH | [196] |
| Moso bamboo ** | PFX, OTC, CTC degradation | Oxidation PDS activation | OCFRs CCFRs-O CCFRs | g2 > 2.0040 2.0030 < g3 < 2.0040 g1 < 2.0030 | ●OH SO$_4$●$^-$ | [197] |
| Bamboo chips | PCB28 degradation | Electron transfer * Oxidation * | PFRs * | | ●OH * | [198] |
| Bamboo | SMX, TOC degradation | Electron transfer Oxidation PDS activation | PFRs | | ●OH, SO$_4$●$^-$ ●O$_2$ | [199] |

* Refers to degradation of PCB28 adsorbed on BBC obtained at low temperature; ** N-doped (urea), S-doped (Na$_2$S$_2$O$_8$), and NS-doped (thiourea); OCFRs = oxygen-centered free radicals; CCFRs-O = carbon-centered radicals with oxygen atoms; CCFRs = carbon-centered free radicals; PFX = pefloxacin; OTC = oxytetracycline; CTC = chloro-tetracycline; PCB28 = 2,4,4′-trichlorobiphenyl; PDS = persulfate.

Briefly, Huang et al., to assess the non-negligible role of biomass types and their compositions on the formation of PFRs in the related BC, selected three different biomass feedstocks, including bamboo [196]. Biomass was pyrolyzed in a tube furnace for 2 h at a temperature of 500 °C, and BBC was stored in an anaerobic glovebox purged with $N_2$ to avoid the effects of oxygen molecules on the PFR concentrations contained therein. The as-obtained BBC was tested in a BBC/$H_2O_2$ Fenton-like system to catalyze the oxidative degradation of tetracycline, evidencing high degradation efficiency by an electron transfer pathway via ROS production. The authors demonstrated that the catalytic property of BBC strongly depended on the amounts of PFRs [196]. The second and last experimental work found in literature concerning BBC-related PFRs was that by Zhang et al., who prepared N-doped (NBC), S-doped (SBC), and NS-doped (NSBC) BBC using Moso bamboo [197]. Bamboo biomass was pyrolyzed at temperatures in the range of 300–700 °C, giving the highest concentration of PFRs at 500 °C. The as-prepared BBCs were used to activate persulfate to produce both radical ($SO_4^{\bullet-}$, $\bullet OH$, and $O_2^{\bullet-}$) and non-radical ($^1O_2$) species, which catalyzed the degradation of different antibiotics. Particularly, both EPFRs enhanced by the N-or S-doping, and non-radical species catalyzed the degradation of different antibiotics, including PFX, OTC, and CTC [197]. More recently, the photodegradation processes under simulated solar illuminations of 2,4,4′-trichlorobiphenyl (PCB28) adsorbed on biochar colloids (BCCs) released from bulk BBCs were studied by Wang et al. as a function of pyrolysis temperature (300, 500, or 700 °C) [198]. It was demonstrated that PCB28 adsorbed on the low-temperature BCCs degraded mainly by accepting electrons from the BBC-associated PFRs. Such electron transfer led to PCB28 dechlorination, followed by ring-opening oxidation through hydroxyl radical attack, ultimately resulting in progressive mineralization [198]. Otherwise, singlet oxygen non-radical mechanisms caused preferential ring opening of adsorbed PCB28 on the high-temperature BCCs, preceding dichlorination [198]. Lignocellulosic bamboo biomass was used as raw material to prepare Fe-N-BBC for persulfate activation and sulfamethoxazole (SMX) and TOC degradation [199]. BBCs were capable of activating peroxydisulfate (PDS) and degraded SMX and TOC, mainly via the oxidative action of four active species, such as $\bullet OH$, $SO_4^{\bullet-}$, $\bullet O_2$, and $^1O_2$ [199]. Based on these data, if the research on BBC is still limited compared to that on BC, the research studying the possible presence of PFRs in BBC, their characteristics, their possible application, their environmental transformation, and above all, their possible toxicity to humans and the environment is even more minimal (Figure 8).

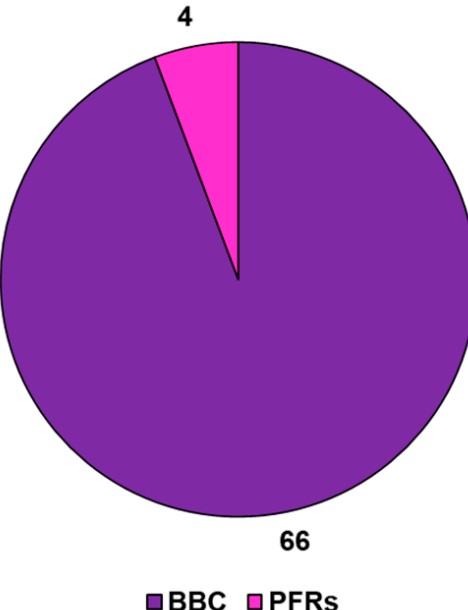

**Figure 8.** PFR-mediated BBC applications vs. BBC applications according to data reported in Tables 6 and 7. Numbers refer to the existing publications reporting that specific application.

### 5. Conclusions and Future Perspectives

In this review, the different applications of biochar (BC) and activated BC obtained by the thermal decomposition of several bamboo species (BBC and ABBC) have been schematically highlighted and extensively discussed. *Bambusa vulgaris*, *Gigantochloa albociliata*, Moso bamboo (*Phyllostachys edulis*), *Dendrocalamus giganteus*, *Bamboo Dendrocatiflorus Munro*, dry bamboo stalks, *Phyllostachys virdiglaucesons*, etc. are the most used bamboo species for developing BBC and ABBC. The interest in bamboo as a source for producing BC is quite young and still too limited, as evidenced by the lower number of existing experimental works concerning BBC compared to those concerning the production of BC from other biomasses. Anyway, BBC has revealed to be promising as an electromagnetic wave absorber, for constructing electrodes to be used in dye-sensitized solar cells, and for water and soil purification by absorption of organic xenobiotics and inorganic pollutants. Additionally, BBC has been helpful for the release of nutrients and gas remediation, and it has been largely demonstrated that BBC positively affects soil properties and greenhouse gas emissions, thus representing a nonpareil natural gift source for BC. Anyway, in our perspective, the more extensive use of bamboo and BBC in the urban, green building, and gardening industries to increase the nation's economic aspects and reduce ecotoxicity, the improvement of the properties of BC developable from bamboo, and more in-depth knowledge about the possible existence of BBC-associated PFRs is mandatory. In this context, the main scope of this review was to support increasing production of BBC and ABBC and to stimulate further studies about their possible applications, thus enlarging the current knowledge about these materials and allowing their more rational, safer, and optimized application, especially in the removal of xenobiotics.

**Author Contributions:** Authors contributed equally to this work. All authors have read and agreed to the published version of the manuscript.

**Funding:** This work was funded by Fundação Carlos Chagas Filho Amparo à Pesquisa do Estado do Rio de Janeiro (FAPERJ project n.SEI-260003/001227/2020), the European Union—*Next Generation EU*-the Italian Ministry of University (MUR) (PRIN2022, project n. 2022JM3LZ3_(SUST-CARB)).

**Institutional Review Board Statement:** Not applicable.

**Informed Consent Statement:** Not applicable.

**Data Availability Statement:** Data and information reported in this work have been obtained by literature using Scopus, PubMed, and Scifinder databases.

**Conflicts of Interest:** The authors declare no conflicts of interest.

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
