# Peer review of "Bamboo-Based Biochar: A Still Too Little-Studied Black Gold and Its Current Applications"

_jox, doi:10.3390/jox14010026_

Round 1

Reviewer 1 Report

Comments and Suggestions for Authors

In their work, the authors made a thorough review of the use of bamboo biochar for the removal of xenobiotics. The production and use of biochar has recently become the focus of interest for many scientists. Therefore, there are many reviews in the scientific literature regarding the production, properties and use of this product in pollution removal processes. In their review, the authors focused on bamboo biochar, which constitutes an important contribution to the development of research on this green sorbent. This work will be a source of valuable information for subsequent researchers. The reviewed article has a good layout, is well prepared and written.

Author Response

In their work, the authors made a thorough review of the use of bamboo biochar for the removal of xenobiotics. The production and use of biochar has recently become the focus of interest for many scientists. Therefore, there are many reviews in the scientific literature regarding the production, properties and use of this product in pollution removal processes. In their review, the authors focused on bamboo biochar, which constitutes an important contribution to the development of research on this green sorbent. This work will be a source of valuable information for subsequent researchers. The reviewed article has a good layout, is well prepared and written.

We thank the Reviewer for his/her positive comments.  

Reviewer 2 Report

Comments and Suggestions for Authors

The paper deals with the important topic of biochar as a material with great importance in recent years. I read the submitted manuscript with great interest. The manuscript's title correctly describes the chosen research topic.  It is structured as a review article that aims to make the state of the art of biochar production and possible usage. The depth of study is demonstrated with the number of references – is enough. Relevance – with the availability of research in recent years – is enough. Interest – with the availability of research by scientists from different countries - is not enough. In Table 1, is there an application in the literature for the last 4 (rice straw, food manure...)

From my point of view, the Conclusion section is formed incorrectly. Conclusion – is a summary of the authors’ study without repetition, and adding new results. 

Comments on the Quality of English Language

Although the authors throughout the paper have shown their ability to demonstrate their knowledge of the field in fluent English, small corrections are necessary.

Author Response

The paper deals with the important topic of biochar as a material with great importance in recent years. I read the submitted manuscript with great interest. The manuscript's title correctly describes the chosen research topic.  It is structured as a review article that aims to make the state of the art of biochar production and possible usage. The depth of study is demonstrated with the number of references – is enough. Relevance – with the availability of research in recent years – is enough. Interest – with the availability of research by scientists from different countries - is not enough. In Table 1, is there an application in the literature for the last 4 (rice straw, food manure...)

We thank a lot the Reviewer for his/her comments, and mainly for his/her final answer that enabled us to improve Table 1 and related text, thus making it clearer. In fact, as written in the Table title, the reported applications in column third are “general” and do not refer particularly to biochar (BC) deriving from a specific biomass among those in first column but refer to BC in general. Better, in the first column we have reported the sources (biomasses) that are commonly used to produce BC, with the related reference (second column), while in the third column we have reported the most common applications of BCs from different biomasses. Anyway, the Table has been modified to avoid confusion and a sentence to clarify the Table structure has been inserted before Table 1. Please, see lines 104-108 and the modified Table 1.

From my point of view, the Conclusion section is formed incorrectly. Conclusion – is a summary of the authors’ study without repetition, and adding new results.

The conclusion section has been modified according to the precious suggestions of the Reviewer. Particularly, repetitions have been removed, while Figure 7 and 8 (original version) have been removed and moved in the main text (revised version) as Figure 6 and 8.

Comments on the Quality of English Language

Although the authors throughout the paper have shown their ability to demonstrate their knowledge of the field in fluent English, small corrections are necessary.

We thank the Reviewer for his/her precious comment. The manuscript has been revised by our colleague Professor Deirdre Kantz, English mother tongue and teaching English at the University of Genoa and Pavia.

Reviewer 3 Report

Comments and Suggestions for Authors

 The review paper by Alfei and Pandoli summarizes the knowledge on biochar making from bamboo and its utilization to remove xenobiotics compounds.

1) The title is attractive certainly, using the terms "black gold" but the applications go beyond xenobiotics removal which makes the content quite misleading. This should be fixed, either correct the title, or focus on xenobiotics removal.

2) Title: "...still too little studied", but when this Reviewer checked Google Scholar using the keywords "bamboo" and "biochar", the search returned many papers in 2024. This should be fixed.

There is only on reference dated 2024 whereas too many appeared on babmboo biochar in this year. This must be fixed. The authors must cite much more bamboo biochar papers from 2023-2024.

3) In light of the additional references, the authors should edit the text.

4) Reference 4 should not be in the list; either cite a preprint, or remove. If the paper is not published, no reader will be able to access it.

5) Figure 1 seems to have an error. The biochar yield at 750-1000 °C should be lower than 37 %. 

The authors should check Figure 1 published in Catalysts 2022, 12, 362. 

6) The discussion of Table 5 seems incorrect when the authors compare O/C ratios. The authors write "The lowest O/C ...(0.96)..." but Table 5 reports an even lower value of 0.76 for forest residue. This is confusing.

7) The paper is on xenobiotics removal but Table 6 encompasses many applications of Bamboo biochar. Either the article title should be corrected, or the applications focussed on xenobiotics removal.

8) Figure 7: The authors mention biochar-based catalysts, they should reports the categories of catalysts: biochar could serve as catalyst for example as acid catalyst (Energy Convers. Manag. 2018, 163, 59–65.), but also when biochar could serve as a support for metallic nanocatalysts.in organic chemistry (Current Opinion in Green and Sustainable Chemistry 2022, 38:100713), dye removal and other processes 

(Lopes & Astruc https://doi.org/10.1016/j.ccr.2020.213585 ; 

Tang et al https://doi.org/10.1016/j.envres.2023.116232 ); 

9) The main issue in this paper is the lack of possible mechanisms of action of the biochar. How removal proceed ? By decomposition, mineralization  or adsorption, and in any case indicate what happens at the molecular level.

Summary: the review merits to be published but after major revisions.

Comments on the Quality of English Language

Generally, the paper reads very well.

Author Response

The review paper by Alfei and Pandoli summarizes the knowledge on biochar making from bamboo and its utilization to remove xenobiotics compounds.

1) The title is attractive certainly, using the terms "black gold" but the applications go beyond xenobiotics removal which makes the content quite misleading. This should be fixed, either correct the title, or focus on xenobiotics removal.

We thank the Reviewer for his/her comment. The title has been modified (line 3).

2) Title: "...still too little studied", but when this Reviewer checked Google Scholar using the keywords "bamboo" and "biochar", the search returned many papers in 2024. This should be fixed.

We thank the Reviewer for his/her comment and understand his/her opinion. Anyway, we would like to make kindly note to the Reviewer, that this review has been written at the early beginning of 2024, when the works cited by the Reviewer (now published) were not still present in literature. Unfortunately, due to a very long revision process, data and information reported in the manuscript result now not updated. Accordingly, we have revised all the manuscript and updated the references including additional references on both biochar and bamboo derived biochar dated 2023 and 2024. Figure 4, 5, 6, 7 and 8 have been updated including also the works published in later 2023 and 2024 (so far). Similarly, Table 7 and Table 8 and the related discussion in the main text have been updated reporting the recent new findings. Concerning the Title, we kindly ask the Reviewer to accept it in the present modified form, since (as explained in the text), we do not mean that bamboo biochar is not studied but that it is still too little studied compared to BC derived from other biomasses.  

There is only on reference dated 2024 whereas too many appeared on babmboo biochar in this year. This must be fixed. The authors must cite much more bamboo biochar papers from 2023-2024.

The request of the Reviewer has been satisfied. Please, consider the works newly added in Table 6 and 7 and in the related discussions comprised in lines 420-469 and 699-715.

3) In light of the additional references, the authors should edit the text.

The text has been edited as above reported.

4) Reference 4 should not be in the list; either cite a preprint, or remove. If the paper is not published, no reader will be able to access it.

As asked, a preprint was cited.

5) Figure 1 seems to have an error. The biochar yield at 750-1000 °C should be lower than 37 %.

The authors should check Figure 1 published in Catalysts 2022, 12, 362.

Figure 1 has been corrected.

6) The discussion of Table 5 seems incorrect when the authors compare O/C ratios. The authors write "The lowest O/C ...(0.96)..." but Table 5 reports an even lower value of 0.76 for forest residue. This is confusing.

The comparison reported in the paper is only between bamboo and wood. Anyway, for better clarity the sentence has been reformulated. Please, see lines 388-392.

7) The paper is on xenobiotics removal but Table 6 encompasses many applications of Bamboo biochar. Either the article title should be corrected, or the applications focussed on xenobiotics removal.

The title has been modified.

8) Figure 7: The authors mention biochar-based catalysts, they should reports the categories of catalysts: biochar could serve as catalyst for example as acid catalyst (Energy Convers. Manag. 2018, 163, 59–65.), but also when biochar could serve as a support for metallic nanocatalysts.in organic chemistry (Current Opinion in Green and Sustainable Chemistry 2022, 38:100713), dye removal and other processes

(Lopes & Astruc https://doi.org/10.1016/j.ccr.2020.213585 ;

Tang et al https://doi.org/10.1016/j.envres.2023.116232 );

In Figure 7 (original manuscript) now Figure 6 with the voice “Catalyst for chemical reactions” we meant the use of BBC to catalyze organic chemical reactions (not including photocatalytic degradations), such as the acidic hydrolysis of cellulose and the pyrolytic production of phenols and aromatic compounds, as already reported in Table 6. Anyway, now it has been specified before the citation of Figure 7 (original version) now Figure 6. Please, see lines 657-664. Additionally, the further category of catalytic esterification has been added together with the relate work both in Table 6 and in the text, as suggested by the Reviewer. Moreover, an additional work on the catalytic hydrolysis of cellulose has been added, both in Table 6 and in the discussion. On other suggestions (Lopes & Astruc https://doi.org/10.1016/j.ccr.2020.2135), a new reference on tetracycline removal has been included in Table 6 and in the discussion, while the work by Wei et al was already included in the original manuscript. Tang et al https://doi.org/10.1016/j.envres.2023.116232 does not concern bamboo-based BC, but among its references, another work on BBC use to absorb CO2 and PM2.5 has been found which has been included in Table 6 and discussion.

9) The main issue in this paper is the lack of possible mechanisms of action of the biochar. How removal proceed? By decomposition, mineralization or adsorption, and in any case indicate what happens at the molecular level.

We make kindly note to the Reviewer, that the main possible mechanisms by which organic and inorganic pollutants could be removed by biochar have been already reported in the original version of our manuscript and precisely in Section 2.2 and subsection 2.2.1. Anyway, in the revision phase more details on the mechanism by which pollutants are removed have been added. As examples, consider lines 536-552, 599-603, 610-616 and 683-715.

 Summary: the review merits to be published but after major revisions.

Comments on the Quality of English Language

Generally, the paper reads very well.

We thank the Reviewer for his/her positive comment.

Round 2

Reviewer 3 Report

Comments and Suggestions for Authors

Accept